# Irisin: A Multifaceted Hormone Bridging Exercise and Disease Pathophysiology

**DOI:** 10.3390/ijms252413480

**Published:** 2024-12-16

**Authors:** Ilaria Paoletti, Roberto Coccurello

**Affiliations:** 1IRCSS Santa Lucia Foundation, European Center for Brain Research, 00143 Rome, Italy; ilariapaoletti93@icloud.com; 2Institute for Complex Systems (ISC), National Research Council (C.N.R.), 00185 Rome, Italy

**Keywords:** irisin, FNDC5, skeletal muscle, physical activity, adipose tissue, insulin resistance, bone pathology, neuroinflammation, neurodegeneration, Alzheimer’s disease

## Abstract

The fibronectin domain-containing protein 5 (FNDC5), or irisin, is an adipo-myokine hormone produced during exercise, which shows therapeutic potential for conditions like metabolic disorders, osteoporosis, sarcopenia, obesity, type 2 diabetes, and neurodegenerative diseases, including Alzheimer’s disease (AD). This review explores its potential across various pathophysiological processes that are often considered independent. Elevated in healthy states but reduced in diseases, irisin improves muscle–adipose communication, insulin sensitivity, and metabolic balance by enhancing mitochondrial function and reducing oxidative stress. It promotes osteogenesis and mitigates bone loss in osteoporosis and sarcopenia. Irisin exhibits anti-inflammatory effects by inhibiting NF-κB signaling and countering insulin resistance. In the brain, it reduces amyloid-β toxicity, inflammation, and oxidative stress, enhancing brain-derived neurotrophic factor (BDNF) signaling, which improves cognition and synaptic health in AD models. It also regulates dopamine pathways, potentially alleviating neuropsychiatric symptoms like depression and apathy. By linking physical activity to systemic health, irisin emphasizes its role in the muscle–bone–brain axis. Its multifaceted benefits highlight its potential as a therapeutic target for AD and related disorders, with applications in prevention, in treatment, and as a complement to exercise strategies.

## 1. Introduction

Irisin, first identified as a protein encoded downstream of the transcriptional peroxisome proliferator-activated receptor gamma (PPAR-γ) co-activator-1 α (PGC1-α), has become almost ubiquitous with muscle physiology, obesity, metabolic syndrome, and diabetes, as well as neurodegenerative diseases. It was first discovered as a product of exercise and muscle activity/contraction [1]. In an attempt to account for the health benefits induced by physical activity on metabolic status, in terms of the browning (i.e., conversion) of white adipose tissue (WAT) and its resistance to diet-induced obesity, irisin was designated a hormonal factor or myokine (a polypeptide of 112 amino acids) cleaved under PGC1-α transcriptional control from the transmembrane precursor fibronectin type III domain-containing 5 (FNDC5) [1] in response to exercise. This research foresaw the potential of the discovery, both for the mechanistic link between exercise and metabolic disorders and for the therapeutic possibilities linked to irisin administration.

In the present narrative review, we will illustrate the fascinating journey of this exercise-mimetic hormone across several stations, leading us to different territories, such as obesity, energy metabolism, adipogenesis and metabolic syndrome, muscle pathology, cardiovascular disease, and age-associated diseases (e.g., osteoporosis and osteoarthritis), finally landing at the neuroprotective role of irisin in age-associated dementia. Although not exhaustive, this literature review aims to provide a conceptual bridge between the effects of physical and motor exercise and various pathophysiological conditions affecting different organs and systems, such as sarcopenia, osteoporosis, and the risk of type 2 diabetes. The review concludes with a discussion on the potential role of irisin as a disease-modifying factor in Alzheimer’s disease pathogenesis. The primary objective is to address gaps in the research on the therapeutic effects of irisin, which have typically been studied in isolation for different conditions. This review seeks to highlight the shared mechanisms through which irisin exerts beneficial effects. From this perspective, it is hypothesized that irisin is a pleiotropic hormone with unique properties that influence multiple signaling pathways, which are common to various pathologies.

Regarding the methodology used, although this is a narrative review, we followed a systematic approach by identifying and utilizing accessible and reputable sources, databases, and search engines, such as PubMed, Scopus, Embase, Cochrane Library, and Google Scholar. The search strategy employed typical keywords, including “irisin” “irisin and exercise”, “irisin and disease”, “irisin and aging”, “irisin and insulin and diabetes”, and “irisin and brain”. Furthermore, as inclusion criteria, we considered only peer-reviewed journal articles, clinical studies, preclinical studies, and systematic reviews, with no time restrictions.

## 2. Irisin, Metabolic Diseases, and Insulin Resistance

The fact that irisin is involved in the regulation of WAT homeostasis is of the foremost importance for its role in the management of energy balance and whole-body metabolism. Being released essentially by skeletal muscle and, in this respect, being a chemokine/myokine, irisin has also been identified in the WAT secretory repertoire and, in particular, in subcutaneous adipose tissue (SAT) [2,3]. Indeed, not long after its identification in muscle as a product of the *fndc5* gene and described as a hormonal factor mediating the positive effects of exercise on energy expenditure by inducing the browning of subcutaneous adipocytes and increasing uncoupling protein 1 (UCP1) expression [1,4], the expression of the *fndc5* gene in muscle was found to be associated with the expression of the *fndc5* gene (and UCP1) in SAT [2]. The same authors also found an inverse correlation between *fndc5* gene expression in adipose tissue and obesity and a positive association between *fndc5* gene expression and the markers of expression of brown adipose tissue (BAT) [2]. This is an important result to be reported, not only because it extends our knowledge about the different sources of irisin production (i.e., muscle/AT) and some mechanisms underlying the balance between WAT (energy storage) and BAT (energy expenditure) and adipocyte metabolism in obesity but also because it indirectly suggests that this adipo-myokine can act as a “signaling factor”, a messenger for interorgan crosstalk communication involving multiple endocrine organs in an orchestrated fashion, including muscle, AT, and bones. Considering the key importance of muscle–AT bidirectional communication for the dysfunction of energy metabolism and, in particular, for obesity and type 2 diabetes (T2D), it is of key interest that the development of insulin resistance (IR) can be regulated by irisin signaling. As a member of skeletal muscle secretome, irisin is far more largely expressed in muscle than in other tissues, such as the heart and AT [2,5,6]. With skeletal muscle being the major tissue in glucose uptake [7] and the main site for glucose disposal via insulin-dependent mechanisms (responsible for over 80% of postprandial glucose disposal) [8], it is fundamental to explore the function of irisin in relation to IR. Basically, what we generally describe as “IR” is a muscle-dependent deficit, involving the decline in skeletal muscle sensitivity to insulin and insulin-dependent glucose uptake [9].

Several hypotheses have been suggested to account for skeletal muscle IR, and one of them focuses on the increase in intramyocellular fat content as well as the excessive accumulation of lipids in the liver [10]. Excessive lipid accumulation in muscle can generate IR development via diacylglycerol (DAG)/ceramide-induced protein kinase C (PKC) signaling, as demonstrated in healthy volunteers that underwent euglycemic–hyper-insulinemic clamping and an increase in plasma free fatty acid (FFA) levels [11], as well as in high-fat diet (HFD)-fed rats that showed DAG and ceramide accumulation, PKC activation, and IR in both oxidative and glycolytic muscles [12]. PKC activation triggers the serine phosphorylation of the insulin receptor substrate (IRS-1) and the consequent reduction in insulin-dependent glucose uptake in muscle [13]. Interestingly, it has been demonstrated that changes in plasma free fatty acid levels directly increase intramyocellular triglyceride (IMCL-TG) content, and that the increase in IMCL-TG is accompanied by the development of IR [14]. Moreover, the accretion of adiposity and obesity are metabolic disorders associated with reduced irisin secretion, and circulating levels of irisin are found to be particularly reduced in (non-metabolically healthy) obese patients with T2D [15,16]. It is also recognized that physical activity provides an anti-inflammatory potential [17], reducing the risk of metabolic diseases, and that irisin is released from the skeletal muscle in proportion to exercise (especially resistance exercise) and muscle contraction that, per se, upregulate the level of PGC-1α and FNDC5 cleavage [18].

Notably, insulin sensitivity and irisin signaling appear intimately interconnected to each other and, sometimes, in an opposite fashion. Elevated blood irisin levels were found to be positively associated with insulin resistance [19], and reduced insulin clearance was observed in adult individuals in association with higher circulating irisin levels [20]. If an increase in irisin signaling is expected to improve insulin sensitivity, the observed data appear to be contradictory or controversial. On the other hand, such inverse associations between irisin levels and insulin sensitivity, with reduced insulin clearance, have been interpreted as a protective and compensatory mechanism of β-cells for impaired insulin sensitivity [20]. In line with these findings, elevated levels of irisin were found in individuals with metabolic syndrome and elevated body mass index, thus suggesting the existence of the compensatory secretion of irisin from muscle and AT because of the possible development of “irisin resistance” in subjects with metabolic disease [19]. One additional hypothesis accounting for IR in skeletal muscle is linked to the idea of mitochondrial dysfunction, which is based on reports describing defective mitochondrial function and, in particular, reduced mitochondria oxidative capacity, low ATP production, lipid peroxidation, and excessive reactive oxygen species (ROS) in the muscle of subjects showing both T2D and IR [21]. Irisin has been shown to increase oxidative phosphorylation and glucose uptake in C2C12 myoblast cells through the p38-mitogen-activated protein kinase (MAPK)-PGC-1α pathway as well as improving mitochondrial respiration [22]. Another relevant mechanism underlying IR can be attributed to the reduction in protein kinase B (PKB/Akt) phosphorylation induced by DAG and ceramide accumulation [23], which is a mandatory route for the translocation of glucose transporter (GLUT4) to the plasma membrane [24]. Whole-body energy metabolism, especially in muscle, can be regulated by irisin signaling through multiple pathways and, in particular, via the activation of AMPK (5′-AMP-activated protein kinase) [25,26], which is a key sensor for energy homeostasis and changes in energy status [27], including the regulation of lipids and glucose metabolism [28], as well as via the PI3K (phosphoinositide 3-kinase)/PKB/Akt pathway [25]. Many of these mechanisms are specifically highlighted and illustrated in Figure 1. Since AMPK, when activated (e.g., by AICAR), can promote GLUT4 translocation from storage to the plasma membrane, and irisin can directly modulate AMPK activation, it is possible to hypothesize that irisin is able to modulate GLUT4 translocation. Irisin can directly phosphorylate AMPK, as shown by the irisin-induced intracellular depletion of ATP in skeletal muscle cells and the parallel increase in AMPK phosphorylation and glucose uptake [29]. Notably, irisin and AMPK signaling appear reciprocally interconnected. Indeed, not only irisin secretion is involved in AMPK phosphorylation, but exogenous AMPK activation (i.e., phosphorylation) facilitates PGC-1α/FNDC5 mRNA expression in C2C12 myocytes [30]. From this view, defective irisin signaling and, downstream, reduced AMPK phosphorylation are other mechanisms by which irisin can contribute to regulating/promoting insulin sensitivity. An interesting study has disclosed some additional aspects of irisin’s contribution to insulin homeostasis [31]. It is known that adiposity and, in particular, visceral fat accumulation and ectopic fat deposition as well as enlarged adipocytes, the excessive release of free fatty acids (FFAs) and the production of reactive oxygen species (ROS) and pro-inflammatory cytokines, are all critical factors for the establishment of hepatic and muscle IR [32,33]. According to this evidence [31], the administration of recombinant irisin can act on β-cells to evoke insulin synthesis and glucose-stimulated insulin secretion. Of note, irisin was shown to exert a protective action on the apoptotic effects induced by selective saturated fatty acids in β-cells via the Akt (a serine/threonine kinase) and Bcl-2 (an anti-apoptotic protein) signaling pathway. Some of these mechanisms are illustrated in Figure 1. Hence, irisin demonstrated its protective potential as a survival factor secreted by muscle under lipotoxic conditions, providing an adaptive response to IR and defective glucose tolerance.

Moreover, the contribution of irisin in the control of insulin sensitivity extends beyond its regulatory role in skeletal muscle and AT, also involving liver metabolism. Hepatic metabolism is crucial for the maintenance of glucose homeostasis and insulin sensitivity, as demonstrated by the severe hepatic insulin resistance developed in animals that underwent selective insulin receptor knockout [34]. As consequence of defective insulin-stimulated muscle glucose uptake, there is an increase in glucose uptake by the liver and potential glucotoxicity. Within this contest, the stress of endoplasmic reticulum (ER) is a pathological condition affecting different organs, cells, and tissues involved in obesity, such as muscle, AT, and the liver [35,36]. T2D is a very insidious clinical condition where IR and pancreatic β-cell failure coexist. Indeed, increased insulin secretion is a maladaptive compensation for IR, which reduces the capacity of glucose utilization by peripheral tissues, such as the liver, and exacerbate the decline in glucose tolerance [37,38]. The resulting chronic hyperglycemia further aggravates IR, deteriorates β-cell function, and increases hepatic glucose production (via gluconeogenesis), generating liver glucotoxicity and both oxidative and ER stress [39]. By contrast, irisin was shown to reduce gluconeogenesis and induce glycogen synthesis, in particular, via the activation of glycogen synthase (GS) phosphorylation downstream of the activation of the phosphoinositide 3-kinase (PI3K)/Akt/glycogen synthase kinase-3 (GSK3) signaling pathway [40]. ER stress and reactive oxygen species (ROS) are intimately connected pathophysiological processes, so that all the multiple factors responsible for ROS production (e.g., defective calcium homeostasis and lipid oxidation) are also accountable for the impairment of ER homeostasis and unfolded protein response (UPR) [41,42]. In an animal model using L-arginine-induced pancreatitis, it was shown that irisin administration reduced both oxidative stress and ER stress in intestinal cells [43], thus demonstrating the antioxidant potential of irisin as well as its ability to alleviate hepatic injury by increasing mitochondrial biogenesis and upregulating uncoupling protein (UCP) 2 expression [44], which plays a role in ROS reduction and mitochondrial function [45]. Several of these mechanisms are illustrated in Figure 1.

Thus, by acting as the anti-ROS agent and being capable of reestablishing ER homeostasis, irisin can contribute to fighting the development of IR, including hepatic IR. As an additional demonstration of these pro-homeostatic mechanisms and protective potential, irisin was shown to be able to improve hepatic IR induced by high glucose–high insulin conditions in human liver-derived HepG2 cells, by stimulating liver kinase B1 (LKB1) phosphorylation and, downstream, by triggering AMPK phosphorylation and suppressing acetyl-CoA-carboxylase (ACC) activity [46].

## 3. Irisin and Muscle/Bone Homeostasis

A noteworthy finding related to the clinical and functional potential of irisin secretion and signaling in bone fracture healing comes from a study examining changes in irisin blood levels in patients with a bone fracture, both before fracture stabilization and postoperatively during the bone union process [47]. Postoperatively, patients had higher irisin levels compared to preoperative values, and these levels were significantly correlated with the bone union process, contributing to fracture healing [47]. Skeletal muscle and bone are dynamic, plastic, endocrine, and paracrine tissues in tight reciprocal control and communication. On the one hand, skeletal muscle and bone work independently to integrate their function in the musculoskeletal system, but on the other hand, they tightly interact with each other, taking advantage of their anatomical and mechanical association as well as from the growing group of hormonal/humoral signaling factors and messengers (i.e., myokines and osteokines), among which irisin may have a prominent role. Of note, osteoporosis/osteopenia and sarcopenia are the two major pathological conditions affecting, respectively, bone and skeletal muscle physiology that can be also viewed as an unitary and complex syndrome known as osteosarcopenia [48]. The occurrence of muscle mass loss and strength deficits (i.e., sarcopenia) with concomitant low bone mass, the deterioration of bone microarchitecture, and bone frailty (i.e., osteoporosis/osteopenia) is the most convincing evidence for muscle/bone crosstalk.

Although both skeletal muscle and bone tissues possess a wide secretory repertoire of bioactive factors and polypeptides, including, for instance, the osteokines osteocalcin, sclerostin, insulin-like growth factor (IGF-1), fibroblast growth factor-23 (FGF23), and the nuclear factor-kappa B receptor activator ligand (RANKL) [49], or the myokines interleukin 6 (IL-6), interleukin 8 (IL-8), myonectin, myostatin, and fibroblast growth factor 21 (FGF-21) [50], the impact of irisin on muscle–bone crosstalk is worth separate consideration. The interesting mechanism is the link between muscle contraction-derived irisin and the effects of this adipo-myokine in terms of the anabolic and catabolic effects on bone mass and skeletal remodeling. The study of the impact produced by irisin on bone homeostasis allows us to introduce the identification of irisin’s binding receptor as a member of the integrin receptor family, which has been possible by investigating the effects of irisin on osteocytes that were shown to be mediated by binding to the integrin complex α1/β1 and, in particular, with the highest affinity by binding with integrin αV/β5 [51]. The identification of integrin αV/β5 as the primary irisin receptor in osteoblasts leaves open the possibility that other “unknown” receptors mediate the effects of irisin in different tissues. A recent paper has reminded us that irisin’s action is primarily mediated through αV integrin receptors, specifically αV/β5, not only in bone but also in fat and the hippocampus [52]. Moreover, this study has enhanced our understanding of the interaction between polypeptides, such as irisin, and receptors like integrins by identifying the exercise-induced extracellular Hsp90α component, which mediates integrin activation through direct binding to the αV/β5 ectodomain [52]. Regarding its effect on bone metabolism, irisin exhibits intriguing potential for apparently contrasting effects, depending on its mode of action (i.e., frequency) and whether its secretion is continuous or pulsatile. In the study mentioned [51], it was found that irisin can stimulate the expression of sclerostin (a Wnt/β-catenin pathway inhibitor) [53] in osteocytes, which in turn, inhibits osteoprotegerin secretion by osteoblasts. This disinhibits the receptor activator of nuclear factor kappa-B ligand (RANKL), the primary factor responsible for osteoclastogenesis [54]. These results appear controversial with regard to the idea that irisin may possess antiresorptive features in bone therapy. However, it should be noted that osteoblast differentiation was enhanced in vitro by irisin-secreting primary myoblasts from mice that were exercised in running wheels for 3 weeks [55]. The exogenous administration of recombinant irisin for 4 weeks during hind-limb unloading in mice has been shown to prevent cortical and, to some extent, trabecular bone mineral density (BMD) loss. Additionally, recombinant irisin administration after the hind-limb suspension period helps recover BMD by promoting the differentiation of bone marrow stromal cells into osteoblasts [56]. Additionally, the cumulative non-continuous once-a-week administration of recombinant irisin was shown to promote osteoblastic-mediated new bone formation via parallel sclerostin downregulation [57]. In agreement, the intermittent administration of recombinant irisin produced the downregulation of sclerostin expression in MLO-Y4 osteocyte-like cells, via an irisin-mediated increase in the activating transcription factor 4 (ATF4) mRNA expression [58]. These studies [57,58] provide excellent examples of irisin exerting inhibitory activity on sclerostin expression, in spite of the apparent controversial result reported by Kim and colleagues [51] in which the irisin-mediated increase in sclerostin is accompanied by osteocytic survival, while the deletion of irisin’s gene inhibited osteocytic osteolysis. Moreover, concerning the irisin-induced activation of ATF4, it should be noted that ATF4 promotes bone marrow mesenchymal stem cell differentiation into osteoblasts, together with the inhibition of osteoclast production [59]. Hence, the irisin-mediated activation of ATF4 can play a key role in osteogenesis and bone formation. It is undoubtedly of interest that serum irisin levels and the risk of osteoporosis are found to be inversely associated, while irisin and BMD are found to be positively correlated [60], and that decreased serum irisin levels in postmenopausal women have been repeatedly reported [61,62]. Further and robust evidence that BMD and bone metabolism (i.e., mineralization) can be regulated by irisin signaling was obtained in FNDC5/irisin deficient mice, which showed reduced BMD and irregular bone development attributable to a defective program of osteoblast-associated gene expression (decreased osteoblastogenesis) and concomitant increased osteoclastogenesis and the expression of osteoclast-associated genes [63]. An interesting additional finding of this study is that exercise-induced weight loss is reduced in FNDC5/irisin knockout (KO) mice. However, recombinant irisin administration during exercise helps restore the balance between physical activity and thermogenesis, along with an increase in the expression of gene markers associated with adipose tissue (AT) browning [63]. Adiposity is a state of chronic low-grade inflammatory status, where the production/secretion of pro-inflammatory cytokines (e.g., TNF-α and IL-6) affect RANK/RANKL concentrations and the osteoprotegerin pathway and, thus, the osteoclastogenesis and bone resorptive activity [64]. The strong connection between obesity and bone mineral density (BMD) loss has made physical exercise one of the best preventive and therapeutic options for maintaining bone health. In a rat model of high-fat diet (HFD)-induced bone mass and strength loss (i.e., impaired bone accrual), an 8-week swimming exercise protocol was administered to investigate the bone’s adaptive response to physical activity [65]. The authors found that swimming exercise evoked an increase in circulating irisin levels and both PGC-1α and FNDC5 in the femur with beneficial effects on bone metabolism, such as a decrease in bone resorption factors (e.g., C-telopeptide of type I collagen (CTX-1) and interleukin (IL)-1) and body fat [65]. With regard to the signaling pathways involved into irisin’s effects on bone homeostasis, there is evidence that irisin can inhibit SOST/sclerostin [57,58], an inhibitor of the canonical Wnt/β-catenin signaling pathway [53], which is responsible for bone formation and the regulation of osteoblast activity (i.e., osteoblast differentiation and osteocyte formation) [66]. Several of these mechanisms are illustrated in Figure 2. In particular, SOST/sclerostin competes and interacts with the Wnt low-density lipoprotein receptor-related protein 5/6 (LRP5/6), thus blocking the activation of Wnt/β-catenin signaling [53]. Interestingly, motor exercise mimetics, such as mechanical loading, have been reported to produce downregulatory effects on the expression of SOST/sclerostin in osteocytes [67], thus evidencing the powerful crosstalk between exercise-induced mechanical loading and the inhibition of Wnt/LRP5/6 signaling.

### 3.1. Autophagy as an Additional Mechanism in Irisin Signaling

Autophagy is another important mechanism by which irisin can promote its pleiotropic effects [68], including those on bone formation and osteogenesis by the autophagy machinery-mediated activation of the Wnt/b-catenin signaling pathway [69]. Thus, by harnessing the potential of bone marrow mesenchymal stem cells (BMSCs) to differentiate into osteoblasts, these authors provide evidence that irisin can upregulate autophagy. This occurs through the increased expression of autophagy-related 5 (Atg5) and microtubule-associated protein light chain 3 (LC3)-I/II in BMSCs, particularly by promoting the formation of the Atg12-Atg5-Atg16L complex and activating the Wnt/β-catenin signaling pathway downstream of the activation of autophagy [69]. In other words, the activation of the Wnt signaling pathway inhibits the differentiation of BMSCs into adipocytes, redirecting their differentiation toward osteoblasts. Irisin contributes to this process through an autophagy-associated mechanism. Additionally, defective autophagy was identified as a key mechanism underlying increased hepatic lipogenesis, decreased fatty acid oxidation, and the severe hepatosteatosis observed in FNDC5 null mice [70]. Figure 2 depicts several of these mechanisms. Specifically, it is known that AMPK activation (e.g., in starving conditions and energy deprivation) can trigger autophagy by the inhibition of the mammalian target of rapamycin (mTOR) signaling and subsequent repression of its functional mTOR complex 1 (mTORC1) [71]. The idea that FNDC5 can trigger autophagy via the AMPK/mTOR signaling pathway is further corroborated by the demonstration that the experimental induction of AMPK phosphorylation is able to prevent defective autophagy in FNDC5 null mice, and that mTORC1 inhibition restores autophagy and reduces liver injury in FNDC5 null mice [70]. Considering that osteoblast differentiation can be stimulated by AMPK phosphorylation and autophagy activation [72], it would be interesting to explore whether irisin is able to stimulate osteoblastogenesis through the activation of the AMPK/mTOR signaling pathway. On the other hand, there is now in vitro evidence that irisin can promote osteoblastogenesis, as shown both in primary osteoblasts from rats and in murine osteoblastic MC3T3-E1 cells, and that osteogenesis is mediated via the activation of p38 mitogen-activated protein kinase (p-p38 MAPK) and extracellular signal-regulated kinase (ERK) signaling cascade [73]. Another mechanistic link between exercise-induced FNDC5 expression and autophagy machinery has been described in age-associated intervertebral disc degeneration, which is a major cause of low back pain and involves the senescence of nucleus pulposus cells. In this study [74], the exercise-induced increase in FNDC5 in blood circulation was shown to modulate the levels of autophagy in nucleus pulposus cells with beneficial effects on the progression of intervertebral disc degeneration.

### 3.2. Irisin Signaling, Chronic Inflammation, and Insulin in Bone and Muscle Homeostasis

Given the strong connection between systemic chronic inflammation and osteoporosis [75], it is interesting to explore how the anti-inflammatory potential of irisin could help clinically address bone pathologies like osteoporosis. Inflammation and osteoporosis coexist during different medical conditions, such as in the postmenopausal period [76], as well as emblematically in chronic and/or autoimmune inflammatory diseases, including rheumatoid arthritis (RA) [77], inflammatory bowel diseases, Crohn’s disease, and ulcerative colitis [78]. Irisin has demonstrated several anti-inflammatory activities. For instance, in AT, mice bearing FNDC5 gene deletion (i.e., FNDC5^−/−^ mice) show IR and macrophage polarization; whereas, exogenous irisin administration is able to reduce AT inflammation via AMPK phosphorylation [79]. Irisin exerts a potent anti-inflammatory action, as evidenced by the ability to decrease NLRP3 inflammasome activation and, in particular, NF-kappaβ (*NF-κB*) signaling in both lipopolysaccharide (LPS)-induced liver injury [80] and endometritis [81]. Interestingly, the macrophage/monocyte-derived pro-inflammatory cytokine TNF-α is responsible for sclerostin upregulation in osteocytes via NF-κB signaling and, therefore, responsible for bone loss [82]. Moreover, there is robust evidence that the inhibition of NF-κB signaling may increase BMD by promoting osteoblasts activity and antiresorptive effects [83,84]. The robust anti-inflammatory potential of irisin, especially via its ability to reduce NF-κB levels via AMPK phosphorylation, further supports the idea that the potentiation of irisin signaling may have beneficial effects as an adjuvant therapy against osteoporosis. Several of these mechanisms are illustrated in Figure 2.

As above described, osteoporosis is often associated with muscle atrophy and muscle weakness, and for this reason, osteosarcopenia represents a syndrome that finely epitomizes the tight muscle/bone interorgan communication. Notably, both osteoporosis and sarcopenia can also be identified in obese patients with visceral adiposity, as described in osteosarcopenic obesity [85], including patients in which low serum irisin levels have been suggested as a possible early diagnostic marker for Cushing’s disease [86]. However, the expression levels of irisin have also been found reduced in age-associated sarcopenia independently from obesity and osteoporosis, as well as exacerbated muscle wasting being reported in FNDC5 KO aging mice [87]. The same study also reported that the exogenous administration of recombinant irisin can improve sarcopenia both in aged and aging mice, thus supporting the idea that irisin might be used to fight age-associated muscle wasting. Since sarcopenia can be found to be associated with IR, irisin signaling can be of key importance as the hormonal messenger linking liver metabolism (e.g., see the previous paragraph), muscle, and bone homeostasis. The abnormal infiltration of lipids in skeletal muscle and, in particular, intramuscular and intramyocellular lipid deposition are responsible for lipotoxicity, which is considered a primary mechanism in myosteatosis-associated inflammation and IR development [88]. Interestingly, β-cell function (i.e., HOMA- β) was found to be negatively associated with osteoporosis when participants in the study had a low IR index (i.e., HOMA-IR), but HOMA-IR was found to be positively associated with osteoporosis in cases of higher HOMA-β, thus indicating the direct relationship between IR development, the deterioration of β-cell function in the lower BMD, and osteoporosis [89]. Hence, insulin can also be an anabolic agent in bone metabolism. Indeed, mice lacking insulin receptors on osteoblasts show a reduction in bone formation and bone mass, and the in vitro proliferation and differentiation of osteoblasts are reduced by the lack of insulin receptors [90]. As a whole, insulin promotes pro-osteoblastic mechanisms and bone formation, while insulin deficiency is accompanied to alterations of bone microarchitecture [91]. One of the mechanisms by which irisin can increase glucose disposal and insulin sensitivity is through the upregulation of GLUT4 downstream of AMPK activation [29,92] as well as via the activation of the AMPK-mediated p38 MAPK signaling pathway [22,29], which is the same pathway mediating osteogenesis in murine osteoblastic MC3T3-E1 cells [73]. Similarly to insulin, irisin can promote GLUT4 translocation in muscle cells, while irisin was not found to stimulate glucose uptake that it is not entirely dependent on GLUT4 translocation [29]. Notably, in primary cultures from mouse osteoblasts, not only has GLUT4 been found to be essential for glucose uptake but its expression was found to increase as a function of insulin-stimulated glucose uptake [93]. In the same study, the use of mice KO for GLUT4 in osteoblasts and osteocytes also discloses the existence of peripheral adiposity, hyperinsulinemia, and reduced insulin sensitivity, thus revealing the functional relationship between alterations of glucose clearance or bone glucose use and whole-body changes in glucose homeostasis [93]. Together, there is, therefore, evidence that irisin can exhibit synergistic anabolic effects together with other anabolic agents, such as insulin in the control of bone and muscle homeostasis. Figure 2 illustrates several of these mechanisms.

With regard to irisin’s ability to fight lipotoxicity-induced IR in myocytes [31], an interesting point is illustrated by its capacity to increase insulin-stimulated glucose utilization using cardiomyocyte cells in a model of palmitic acid-induced IR [94]. Similarly to the antiapoptotic effects found in β-cells and glucose-stimulated insulin secretion described after recombinant irisin administration [31], this study [94] demonstrated that irisin can contribute to improving insulin signaling in cardiomyocytes via the activation of the PI3K-Akt pathway, thus providing high benefits in terms of the possible prevention of metabolic alterations, cardiovascular diseases (e.g., myocardial infarction), heart failure, and the general deterioration of cardiac function. Although, recently, the role of irisin signaling in cardiovascular diseases has received meticulous attention [95], the impact that irisin may have on the control of blood pressure and in the etiology of hypertension warrants a particular mention. It is known that the activation of the sympathetic nervous system is essential in the pathogenesis of hypertension. Nevertheless, understanding the pathogenesis of hypertension can be elusive because of the many mechanisms recruited, from those involving renal and vascular processes [96] to those involving immune, inflammatory, and brain mechanisms [97,98]. Multiple brain nuclei in the hypothalamus, forebrain, and brainstem contribute to integrate neurohumoral signals from the periphery for the regulation of blood pressure and osmotic homeostasis [98]. Among the hypothalamic nuclei, the paraventricular nucleus (PVN) has a major importance in the regulation of blood pressure, playing a key integrative role for cardiovascular function, which is subserved by its neuroendocrine system, including several neurotransmitters and neurohumoral factors, such as vasopressin and oxytocin [99]. Of interest, there is, for instance, the ability of intravenous irisin administration to activate the hypothalamic nuclear factor E2-related factor-2 (Nrf2) and lower blood pressure by reducing plasma noradrenaline levels and the activation of PVN in spontaneously hypertensive rats [100]. Within this context, the point of particular interest stands in the irisin’s ability to activate hypothalamic circuitries and modulate PVN output, also reducing oxidative stress and neuroinflammation associated with pro-inflammatory cytokine (e.g., IL-1β) [100]. Such antihypertensive and anti-inflammatory activity in the PVN led us directly to the role of irisin in the brain and to the crosstalk between the muscle/bone axis and the brain.

## 4. Irisin, Neurodegeneration, and Alzheimer’s Disease

Age-associated dementia (aged ≥ 65 years) is the most prevalent worldwide neurodegenerative disease, showing a rising number of afflicted individuals, which is estimated to grow from about 57 million in 2019 to about 150 million by 2050 [101]. In turn, Alzheimer’s disease (AD) is the predominant form of age-associated dementia whose global estimated incidence is around 32, 69, and 315 millions of people with AD dementia, prodromal AD, and preclinical AD, respectively [102]. Since, from the identification of the “typical” neuropathological hallmarks of AD, namely, the aggregation of toxic species amyloid-β (Aβ) and the formation of Aβ plaques, together with dystrophic neurites and the development of neurofibrillary tangles (NFTs) of hyperphosphorylated tau protein (abnormal tau tangles) [103], the idea of Aβ accumulation as a key element of gradual cognitive decline and neuronal death has been the mainstay of the amyloid cascade hypothesis [104,105]. The consistency of the amyloid cascade hypothesis has been recently discussed elsewhere [105,106], and within this framework, our main point is neither to discuss the idea that Aβ accumulation is the primary trigger for other pathogenetic events, such as hyperphosphorylated tau and dementia, nor to examine whether Aβ is the major target for the development of AD therapy.

In parallel, there is robust evidence that irisin signaling can interfere with several neuroinflammatory processes, which are pathogenetic events in AD progression. Of key importance, irisin has been detected (i.e., its immunoreactivity) not only in muscle, cardiomyocytes, the liver, and the pancreas but also in neurons and, in particular, has been labeled in the Purkinje cells of the cerebellum [107], hypothalamus [108], and spinal cord [109]. An intriguing “proof of concept” demonstrating irisin’s potential to fight neuroinflammation and its relevance to Alzheimer’s disease (AD) pathogenesis has been provided through studies investigating Aβ neurotoxicity in primary astrocyte and neuron cell cultures [110]. Irisin pretreatment for 12 h of cultured astrocyte exposed to Aβ was shown to exert a marked neuroprotection against Aβ neurotoxicity (e.g., cell viability loss) and reduce IL-1β and IL-6 release. Considering the role of pro-inflammatory cytokines, such as IL-1β and NF-κB, in inducing cyclooxygenase 2 (COX-2) expression [111], including in the brain [112,113], the study demonstrated that irisin treatment effectively inhibited the expression of COX-2 and NF-κB in astrocytes exposed to the Aβ challenge [110]. Consistently, irisin can protect against neuronal injury, as in the case of cerebral ischemia induced by the middle cerebral artery occlusion (MCAO) in mice [114]. This study demonstrated that recombinant irisin administration can reduce brain infarction, brain edema, microglial activation, neutrophil infiltration, oxidative stress, and the expression of TNF-α and IL-6 [114]. These effects are mediated through the activation of the ERK1/2 and Akt signaling pathways, similarly observed in the previously described protective action against apoptosis induced by saturated fatty acids in β-cells [31]. The neuroprotective mechanism linked to exercise-induced irisin production [114] supports the connection between the beneficial effects of physical activity and disease-modifying therapies in Alzheimer’s disease (AD) [115]. It also reinforces the concept of irisin as a key mediator in the muscle–brain axis, bridging exercise-induced muscle contraction, myokine secretion, and brain health. Additionally, in the same MCAO model, irisin has been shown to inhibit NF-κB activation and downregulate the Toll-like receptor 4 (TLR4) signaling pathway in a MyD88-dependent manner [116]. In particular, irisin pretreatment reduced infarct size and neuronal damage induced by MCAO in rats by decreasing the NF-κB activation and TLR4/MyD88 mRNA expression and, thus, the overall inflammatory response [116]. TLR4 is a pattern recognition receptor (PRR), highly expressed by innate immune cells, such as microglia, the resident immune cells of the brain [117]. Microglial cells, with their plastic ability to continuously survey and scan the surrounding environment using motile processes, shape synapses and play a dual role in immune defense and the maintenance of CNS homeostasis [117]. However, the activation of TLR4 is considered a critical trigger for neural loss and neurodegeneration [118]. The TLR4 can recognize pathogen-associated molecular patterns (PAMPs) (e.g., bacteria, virus, or fungi) as well as damage-associated molecular patterns (DAMPs) released after tissue damage [119], and its expression, together with the production of pro-inflammatory cytokines, has been found to be increased in the frontal cortex of AD patients [120] and in the AD-like APP/PS1 mouse model with microglial cell activation, neuronal loss, and exacerbated cognitive deficits [121].

Thus, irisin can exert powerful anti-inflammatory actions, as evidenced by its effects on LPS-induced systemic inflammation and macrophage polarization [122]. Following LPS-induced sepsis and the resulting shift to pro-inflammatory M1-type macrophage differentiation, irisin administration was found to promote M2 macrophage polarization and the expression of M2 markers, such as IL-10. This process alleviated the severe inflammatory condition, potentially through the upstream increase in JAK2 phosphorylation and STAT6 activation, as well as the activation of peroxisome proliferator-activated receptor gamma (PPAR-γ) and Nrf2. This mechanism is reminiscent of the association between irisin’s anti-hypertensive effects and hypothalamic Nrf2 activation in the paraventricular nucleus (PVN), as discussed in paragraph 3 [100,122]. A similar shifting from M1 towards the M2-like anti-inflammatory phenotype was also observed in the investigation of the changes in microglia morphology after irisin administration, as described in the abovementioned study of neuroprotection in mice subjected to MCAO [114]. Among the different mouse models of neuroinflammation, the intracerebral hemorrhage (ICH) model is a type of brain stroke that can produce intracranial hypertension and secondary brain injury characterized by oxidative stress, neuronal apoptosis, and severe neuroinflammatory response, accompanied by a cascade of events, such as microglia activation, pro-inflammatory cytokine release, and immune cell infiltration [123]. An important aspect of understanding the mechanisms underlying irisin’s neuroprotective potential is the finding that the integrin αV/β5 receptor, previously identified as mediating the effects of irisin in osteoblasts [51], is also highly expressed in microglia [124]. Thus, the expression of the integrin αV/β5 receptor was found to be increased 24 h after the implementation of ICH [125], while its inhibition abolished the positive effects produced by irisin administration on neuroinflammation, in terms of a reduction in brain edema, the inhibition of microglia M1-like polarization, neutrophil infiltration, and neuronal cell death. This provides evidence of neuroprotective potential, mechanistically linked to the increased expression of the integrin αV/β5 receptor, along with enhanced AMPK phosphorylation and the upregulation of anti-apoptotic Bcl-2 protein expression [125]. This is also reminiscent of the protective effect of irisin against apoptosis induced by saturated fatty acids in β-cells [31]. Of note, there is extensive evidence that shifting to M2 polarization can be induced by anti-inflammatory cytokines, such as IL-4 and IL-13 [126], and from this view, irisin seems to act in a cytokine-like fashion. Indeed, not only can irisin evoke microglia M2-like morphology in a cytokine-like manner, but the shift towards the M2 polarization can in turn induce the production of anti-inflammatory cytokines, such as IL-10 (especially after TLR-4 stimulation) [127], as well as transforming growth factor (TGF-β) [128], and neurotrophic growth factors, such as brain-derived neurotrophic factor (BDNF) [129] and IGF-1 [130]. Several of these mechanisms are illustrated in Figure 3.

The relationship between neurotrophins, such as nerve-derived growth factor (NGF) and microglia, appears to be bidirectional. Microglia can express the tyrosine-kinase tropomyosin A (TrkA) NGF receptor that, upon its activation (e.g., by the neurosteroid dehydroepiandrosterone), can elicit powerful microglia-mediated anti-inflammatory responses [131]. In turn, NGF via its receptor TrkA located on microglial cells, has been shown to downregulate LPS-induced pro-inflammatory cytokines (TNFα, IL-6, and IL-1β) and nitric oxide production in mouse microglia [132]. Notably, in a study focused on some molecular underpinnings associated with the antidepressant potential of irisin administration [133], it was reported that short-term irisin treatment increased the expression of neutrophins in the hippocampus and prefrontal cortex (PFC) and, in particular, the expression of hippocampal *Ngf* mRNA and *Bdnf* mRNA in the PFC. Physical exercise has a well-recognized defensive role against age-associated neurodegeneration [115], and senescence-accelerated prone mice (SAMP-10 mice) subjected to a program of passive motor exercise showed a selective increase in irisin expression in brain areas, such as the hippocampus and medial prefrontal cortex [134], which play a critical function in AD-associated cognitive decline. Along with the increase in irisin expression in the hippocampus, this study also revealed a parallel upregulation of NGF signaling in the same brain areas, hence contributing to the preservation of neural density and to the reduction in age-associated memory decline [134]. BDNF expression, not only in neurons but also on microglial cells, can play a critical role for neuron survival, neuronal differentiation, neurogenesis, the regulation of long-term potentiation, synaptic transmission, and experience-dependent synaptic plasticity, all processes involved in neurodegeneration [135,136]. The genetic deletion of BDNF from microglia in mice was shown to produce effects similar to those observed after genetic whole-brain microglia depletion [137]. A decrease in learning-induced spine formation, but not spine elimination, was described in mice with selective microglial BDNF deletion [137]. On the one hand, BDNF appears to participate in the regulation of microglial dynamics, and on the other hand, BDNF itself can be considered downstream of exercise-induced irisin expression. The reciprocal crosstalk between irisin and BDNF signaling is important for the understanding of the effects of irisin in neurodegenerative diseases such as AD. For instance, the serum levels of both irisin and BDNF were found to be increased and correlated with motor exercise in elderly women [138]. The most convincing evidence that exercise-induced FNDC5/irisin brain expression is required for increased BDNF has been provided by the demonstration that the forced expression of FNDC5/irisin in cortical neurons is responsible for the increase in BDNF within the brain [139]. This study further showed that the peripheral delivery of FNDC5 via adenoviral vectors can also evoke the expression of *Bdnf* in the hippocampus, supporting the captivating view that circulating FNDC5 may cross the blood–brain barrier and promote the key action of BDNF for hippocampal function and cognitive performance. Both exogenous BDNF administration and an exercise-induced increase in BDNF signaling can reduce the production of toxic Aβ peptides in transgenic AD-like APP/PS1 mice, thus corroborating the idea of a direct involvement of BDNF in non-amyloidogenic APP processing [140]. In a seminal paper, a reduction in the FNDC5/irisin levels in both the hippocampus and cerebrospinal fluid of AD patients, as well as in the brain (i.e., hippocampus and cortex) of different transgenic AD-like mouse models, was shown [141]. In the same study, the knockdown of brain FNDC5/irisin in C57BL/6 mice produced severe memory deficits and the impairment of synaptic plasticity, while exercise (i.e., daily swimming) was able to protect against memory deficits induced by intracerebroventricular amyloid-β oligomer infusion in Swiss mice [141]. Moreover, the neurotoxic effects produced by the intracerebroventricular infusion of Aβ_1-42_ oligomers were found to be responsible for brain BDNF downregulation, and FNDC5 overexpression in neural cell lines was shown to abolish the suppressive effect of Aβ_1-42_ oligomers on BDNF expression [142].

An interesting mechanism hypothesized to explain the relationship between physical exercise, FNDC5/irisin activation, BDNF expression, and the positive effects on cognitive performance focused on data reporting the role of lactic acid as the metabolic messenger linking muscle contraction to the FNDC5/irisin–BDNF axis. It is reported that not only can voluntary exercise increase lactate levels in the hippocampus together with hippocampal BDNF expression and memory formation, but also that the NAD^+^-dependent histone deacetylase silent information regulator 1 (SIRT1) can activate the PGC-1α/FNDC-5 signaling pathway and induce *Bdnf* expression in the hippocampus [143]. In other words, the effects of lactate appear dependent on SIRT1 activation, which, in turn, triggers a SIRT1/PGC1α/FNDC5 pathway, causing a downstream increase in *Bdnf* expression [143]. Several of these mechanisms are depicted in Figure 3.

Notably, AD symptoms are characterized not only by progressive cognitive deterioration but also by the early appearance of a constellation of different neuropsychiatric symptoms (NPSs) [144,145]. Depression and apathy are the most frequent NPSs diagnosed even in mild cognitive impairment or in the early stages of AD, while psychosis, delusions, hallucinations, aggression, and a decline in executive function appear more common as the disease progresses [144,145]. Reduced levels of BDNF have been repeatedly associated with neurodegeneration, and reduced BDNF signaling is correspondingly associated with the damage or deterioration of cognitive functions. Indeed, in a pioneering report, BDNF mRNA was found to be reduced in the hippocampus of AD patients [146]. According to another report [147], in which neuritic plaques within the entorhinal cortex were found inversely associated with *Bdnf* gene expression in the hippocampus, defective BFNF signaling might occur both upstream of the neuropathological hallmarks of AD and downstream of the exacerbation of amyloid plaque load and abnormal tau tangles. Hence, depression is frequently diagnosed in the early stages of AD progression, and at the same time, reduced levels of both BDNF mRNA and serum BDNF levels have been described in people with depression [148]. Between depressive symptoms and brain dopamine homeostasis exists a complex relationship. A dysregulation of dopamine metabolism/signaling is considered a critical factor in depressive symptoms and, above all, for anhedonia symptomatology [149]. Moreover, it should be considered that experimental lesions to the ventral tegmental area (VTA) produce anhedonia and learned helplessness behavior [150], while the stimulation of the VTA can elicit resistance to depression-like behaviors (e.g., social avoidance) [151]. Similarly, the pharmacological increase in dopaminergic tone, for instance by administration with dopamine D2 and D3 (and serotonin 5-HT_1A_) receptor partial agonists, such as aripiprazole, can produce relief from depressive symptoms, especially for treatment-resistant depression [152], and cabergoline administration in rats was shown to relieve depression-like behaviors and potentiate BDNF intracellular signaling in the hippocampus [153]. In turn, a decrease in BDNF levels/signaling is considered a risk factor in depression pathogenesis [154] and a consistent biomarker in mood disorders [155]. Remarkably, not only the mesolimbic dopamine circuit is relevant for depression pathophysiology [156], but early VTA neural loss, dopamine depletion, and reward processing deficits have been found in AD-like mice [157].

Hence, irisin function in the brain can be placed at the intersection between its neuroprotective anti-inflammatory activity, the activation of BDNF signaling, AD pathogenesis, and depression. A direct relationship between physical exercise, an increase in BDNF levels in striatum, and dopamine release in both dorsal and ventral striatum (i.e., nucleus accumbens) have been recently described [158]. Thus, if physical exercise is now acknowledged as an effective and safe therapy for attenuating depression symptoms [159] and a disease-modifying therapy in AD [115], irisin appears the most convincing candidate to mediate these effects. Irisin administration has been shown to recover depressive-like behaviors, such as sucrose preference, by mechanisms linked to the increase in glucose transport and AMPK phosphorylation [160]. Eight weeks of treadmill running was shown to increase brain mitochondrial biogenesis and, in particular, the expression of PGC-1α and SIRT1 mRNA not only in muscle (i.e., soleus) but in several brain regions including the cortex, hippocampus, and midbrain [161]. The increase in FNDC5 expression in midbrain VTA corroborates the role of irisin as a hormonal messenger, able to evoke BDNF expression and, thus, stimulate mesocorticolimbic dopamine function for the regulation of reward-associated processes and motivation. As mentioned earlier [139], the levels of FNDC5 expression in the hippocampus decreased in PGC-1α null mice, providing evidence that, similar to muscle, PGC-1α can also induce irisin expression in the brain. Additionally, the increase in brain-derived neurotrophic factor (BDNF) in the brain is dependent on FNDC5/irisin expression [139]. It has been suggested that irisin-induced BDNF expression and the activation of its high-affinity receptor, tropomyosin-related kinase B (TrkB), can stimulate dopamine 3 (D3) receptor-mediated signaling through the Akt and ERK pathways. Specifically, this signaling enhances the expression of presynaptic D3 receptors in the ventral tegmental area (VTA) [162], which plays a crucial role in fine-tuning dopamine release [163] and in the reward processing of salient information [164]. Further recent evidence from a high-performance liquid chromatography study in mice showed that the daily administration of recombinant irisin for 10 weeks resulted in increased dopamine levels in the nucleus accumbens [165]. Several of these mechanisms are portrayed in Figure 3.

Finally, a recent report has provided robust evidence that chemogenic stimulation inducing dopamine-evoked release from the VTA can trigger a downstream the increase in neprilysin activity [166], which is an enzyme highly expressed in the hippocampus [167] and fundamental for Aβ degradation and the regulation of Aβ levels [168]. Thus, the increase in mesocorticolimbic dopamine release was shown to reduce Aβ deposits and Aβ burden in the prefrontal cortex [166], supporting the view that early dopamine neuronal loss in VTA is an early event in AD pathogenesis [157], and dopamine-based strategies can help in the understanding of AD development and the design of novel therapies. The pleiotropic range of actions associated with irisin/FNDC5 signaling may provide an innovative perspective in this context. By linking physical exercise to neuroprotection, anti-inflammatory activity, increased BDNF signaling, the prevention of dopamine neuron loss [169], antidepressant efficacy, and the potential to enhance reward processing and motivation in AD patients, irisin appears to possess many qualities that warrant further exploration as a therapeutic option in Alzheimer’s disease, both for controlling cognitive decline and managing NPSs.

## 5. Main Limitations in Findings

This section addresses key limitations inherent to both clinical and preclinical studies focused on the role of irisin, without diminishing its therapeutic relevance and innovativeness of this pleiotropic messenger in various pathological conditions.

First of all, it should be mentioned that several studies show a certain degree of measurement inconsistencies, that is, the lack of standardized methods for measuring irisin levels, especially considering its low concentrations in blood. Indeed, different studies use various techniques, such as ELISA and mass spectrometry, which may yield inconsistent results due to differences in sensitivity and specificity, as for the case of the management of coronary heart disease, which is emblematic of the whole problem [169]. Moreover, many studies, and particularly those in the early phases of research, rely on small sample sizes, which affect the robustness and generalizability of the findings. Larger, more dissimilar populations are needed to confirm the results and assess the true impact of irisin across different demographics [169,170,171]. It should also be acknowledged that the effects of irisin are not always consistent across studies. For example, some studies suggest that irisin has a protective effect on metabolic health, while others fail to replicate these findings. This uncertain consensus may complicate the understanding of its role and therapeutic potential [171]; although, in the present survey, we collected a wide-ranging collection of recent literature in which several methodological limitations have been addressed or standardized. There are, however, some additional points of criticism to the studies so far completed. The issue of the limited longitudinal data is undeniably one of them. Most studies focus on short-term effects of irisin, with limited research on its long-term impacts on human health. This makes it difficult to understand the sustained therapeutic benefits or risks of manipulating irisin levels over time. Moreover, the limitation of translatability from preclinical animal models to humans should always be considered. Most of the irisin research is conducted using animal models, particularly rodents, and while these models provide very useful insights, their applicability to humans remains limited. Most research focuses on short-term or cross-sectional data, which does not provide an insight into the long-term impact of irisin on bone health or its potential as a therapeutic target. Thus, more longitudinal studies are needed to assess how irisin’s effects evolve over time.

Altogether, there are significant differences in irisin levels reported across studies, even for the same pathology. Variability arises from differences in the study design, participant characteristics, measurement methods, and sample sizes, which may explain inconsistent findings. For example, some studies report lower irisin levels in individuals with obesity, which is often associated with impaired metabolic function. For example, some studies found significantly lower irisin levels in obese individuals compared to healthy controls, linking this to poor insulin sensitivity and reduced fat burning capacities [2,172]. Other studies, however, have found no significant differences or correlations in irisin levels between obese and nonobese children, suggesting that irisin’s role in obesity may be influenced by other factors, like physical activity or age [173]. Some studies report decreased irisin levels in patients with TD2, correlating lower levels with insulin resistance and poor glucose metabolism [174]. However, irisin may not consistently serve as a reliable biomarker for type 2 diabetes due to significant variability in its levels across patients, influenced by factors such as disease stage, medication use, and individual metabolic conditions. As reported above (see paragraph 4), many studies indicate decreased irisin levels in AD compared to healthy controls, correlating these reductions with neurodegeneration, poor neurogenesis, cognitive decline, and neuroinflammation. However, there are also others report in which no significant changes or even increased irisin levels in certain subgroups of Alzheimer’s patients were found. For example, in a study involving Alzheimer’s patients and healthy age-matched controls, a slight increase in serum irisin levels was reported, specifically in Alzheimer’s patients exhibiting agitation/aggression. This elevation showed a positive correlation with the duration of these neuropsychiatric symptoms [175]. Additionally, serum irisin levels did not correlate with variables such as sex, age, disease duration, cognitive function scores, or the use of medications like acetylcholinesterase inhibitors [175]. Variability could be explained by the different AD stages, coexisting health conditions, and the methods used to measure irisin in the brain versus peripheral fluids. In summary, despite significant variations in irisin levels between healthy and diseased states, its potential for use in the diagnosis and therapy of conditions like obesity, diabetes, osteoporosis, and Alzheimer’s disease remains unchanged. Differences across studies for the same pathology are common and could be attributed to the heterogeneity of the patient populations, experimental conditions, and measurement techniques. Despite these differences, the general consensus suggests that irisin levels are lower in individuals with metabolic, musculoskeletal, and neurodegenerative diseases compared to healthy controls.

## 6. Conclusions and Future Directions

To conclude the discussion on irisin’s role in physiology and pathology, it is important to summarize the unique characteristics of this exercise-mimicking adipo-myokine. Special attention has been given to the beneficial effects of irisin, particularly in relation to osteosarcopenia, insulin resistance, and age-related Alzheimer’s disease, as well as its role in interorgan communication through irisin signaling. Since irisin is considered an exercise mimetic, its beneficial effects should ideally be observed, even in the absence of physical activity. Several studies have demonstrated that recombinant irisin administration can mimic the effects of endurance–strength exercise, highlighting its potential therapeutic value. This aspect enhances irisin’s therapeutic possibilities, reinforcing its role in promoting health. When considering its therapeutic potential in preventing or slowing Alzheimer’s disease progression, irisin’s mechanisms and effects appear to meet the necessary criteria for a disease-modifying agent.

Irisin has shown potential in preventing Alzheimer’s disease (AD)-associated neuronal loss, cognitive decline, synaptic damage, neuroinflammation (e.g., by inhibiting microglia activation, IL-1β expression, and neutrophil infiltration), oxidative stress, and neuropathological changes (e.g., reducing Aβ deposits and NFTs). Additionally, it promotes the expression of neurotrophins, like BDNF, which is critical for neuron survival, synaptic plasticity, and learning and memory processes. Furthermore, irisin’s activation of BDNF signaling is crucial for modulating dopamine release, and its antidepressant effects are important not only for mood disorders but also for managing neuropsychiatric symptoms in AD patients. As noted, increased dopaminergic tone—either through the administration of levodopa (L-DOPA) or the chemogenetic stimulation of dopamine release in the ventral tegmental area—has been shown to degrade Aβ and reduce its aggregation by enhancing neprilysin-dependent Aβ peptide digestion [166]. The tight interconnection between dopamine signaling, irisin, and AD is further suggested not only by the evidence that early neuron loss in the VTA is reported in AD-like mice [157] and the dopamine-evoked neprilysin activity [166] but also by the captivating finding that irisin was able to reduce Aβ pathology using a neprilysin-mediated mechanism [176]. Particularly, it was shown that irisin can act on astrocytes and stimulate an astrocytic-dependent release of neprilysin. Among the different mechanisms previously reported to account for irisin’s neuroprotective potential, there is the expression of the αV/β5 receptor, found both in osteoblasts [51] (paragraph 3) and in microglia [124] (paragraph 4), which contributes to mediate the effects on bone metabolism and anti-inflammatory activity.

By exploring the mechanisms underlying the irisin-evoked ability to decrease Aβ pathology via neprilysin activity, the authors of the study found that the integrin αV/β5 receptor located on astrocytes was responsible for the reduction in Aβ levels [176]. This serves as another extraordinary example of irisin’s pleiotropic action, demonstrating its ability to provide various therapeutic effects through common mechanisms that target different biological systems. In the context of Alzheimer’s disease, this study offers the crucial “missing link” between irisin, dopamine, neprilysin-dependent Aβ degradation, and the neuropathological changes observed in AD. Finally, the role of irisin in neurodegenerative diseases has yet to fully address a critical aspect of AD pathogenesis. Aging, a common risk factor for both AD neuropathology and the decline in brain insulin levels, contributes to the development of insulin resistance [177]. Additionally, a decrease in brain glucose metabolism, or hypometabolism, can often be detected years before an AD diagnosis [178].

In brief, the brain has a limited glucose storage capacity and is, therefore, highly vulnerable to a scarcity in glucose supply [179]. A decrease in glucose metabolism has been observed in aging mice in association with the reduction in GLUT1 expression and cognitive decline [180], which may be dependent on hyperglycemia-induced insulin resistance [181], together with the loss of both GLUT1 and GLUT3 levels reported in the brain of AD patients [182]. Currently, the significant role played by dysfunctional insulin signaling in AD pathogenesis is well established [183], as well as the reduced expression of IRS-1 and insulin-like growth factor 1 (IGF-1) in AD patients [105]. In terms of signaling pathways, and in cases of diabetes, reduced insulin levels and reduced insulin receptor expression are associated with AD and the defective activation of the insulin/IGF-1/PI3K/Akt pathway of intracellular signals, which is a major effector for insulin’s action in the brain [105,184].

An intriguing aspect of brain insulin resistance in AD is the interplay between insulin/p-IRS/PI3K/Akt signaling and the upregulation of BDNF. BDNF not only activates PI3K/Akt signaling but also enhances synaptic plasticity, thereby offering potential cognitive benefits [185]. In a brain insulin-resistant state, both microglia and astrocytes become chronically hyperactivated. This neuroinflammatory condition, characterized by reduced M2-like polarization, leads to a decrease in BDNF levels [186]. On this ground, BDNF delivery into the brain has been imagined as a strategy to improve AD neuropathology. Various methods have been employed to enhance the delivery of BDNF to the brain, including the use of nanoparticles derived from modified liposomes [187], BDNF-releasing encapsulated cell devices in AD-like mice [188], and intranasal delivery to improve cognitive function [189]. Irisin–BDNF crosstalk is another compelling example of irisin’s pleiotropic effects. As discussed earlier, irisin directly influences insulin sensitivity, with regulatory mechanisms extending beyond skeletal muscle and adipose tissue to include liver metabolism. From this perspective, the therapeutic potential of irisin requires further investigation, particularly regarding its role in brain insulin resistance, which may act as an additional pathogenetic factor in Alzheimer’s disease. Dysfunctional IRS-1/2 signaling, resulting from chronic neuroinflammation and the toxicity of Aβ oligomers, is considered a key element in the development of insulin resistance in AD [190,191]. Notably, irisin has been shown to enhance PI3K/AKT insulin signaling and reduce the serine phosphorylation of IRS-1 and IRS-2 [192], which are key molecular mechanisms contributing to insulin resistance [193].

The growing scientific interest in irisin and its role in various metabolic and therapeutic processes can allow for the identification of multiple future research directions. One of these could be focused on the better identification, characterization, and understanding of irisin’s cellular receptor(s), which is a critical step toward unraveling its full biological effects and potential applications in medicine. An interesting field of research that deserves to be developed is the investigation of irisin’s role in different cancers. Indeed, while preliminary studies indicate anti-proliferative effects on certain cancer cell lines, more research is needed to clarify its molecular pathways and therapeutic viability [194]. Tissue regeneration is undoubtedly another field that holds potential. Irisin has shown potential in promoting tissue repair, including bone and muscle regeneration, which could lead to novel treatments for degenerative diseases or injuries [195]. Finally, in the field of biomarker development, irisin holds potential as a biomarker for metabolic health and disease progression, and this aspect requires further validation [196]. Moreover, identifying gaps, such as the lack of long-term human studies on recombinant irisin therapy, is a valuable point for advancing in this field. Irisin, while promising in preclinical studies, faces significant challenges in translating findings to clinical applications. For instance, data could be downsized by the problem of therapeutic formulations and recombinant irisin’s stability, bioavailability, and delivery methods. Long-term studies could address whether sustained delivery systems or periodic dosing is optimal. Thus, closing this gap could significantly enhance our understanding of irisin’s therapeutic potential and aid in developing robust, evidence-based clinical applications.

Another final consideration is about irisin’s potential to enhance insulin signaling, a feature that suggests its relevance in addressing insulin resistance-related conditions, such as AD. Indeed, similar to the therapeutic repurposing of glucagon-like peptide-1 (GLP-1) mimetics and analogs [197], irisin’s ability to enhance insulin signaling suggests it could be a promising candidate for Alzheimer’s disease (AD) therapy. Insulin signaling may represent the intersection of irisin’s pleiotropic effects on health, as demonstrated in cases of muscle mass loss and age-related insulin resistance. Maintaining skeletal muscle mass is a key defense against impaired glucose metabolism and insulin resistance. As thoroughly discussed, irisin plays a crucial role in protecting muscles from atrophy, helping preserve both mass and strength. By preserving skeletal muscle integrity, irisin helps maintain insulin sensitivity across various tissues and organs, including bone. This, in turn, offers neuroprotective effects, potentially providing therapeutic benefits for conditions such as AD.

## Figures and Tables

**Figure 1 ijms-25-13480-f001:**
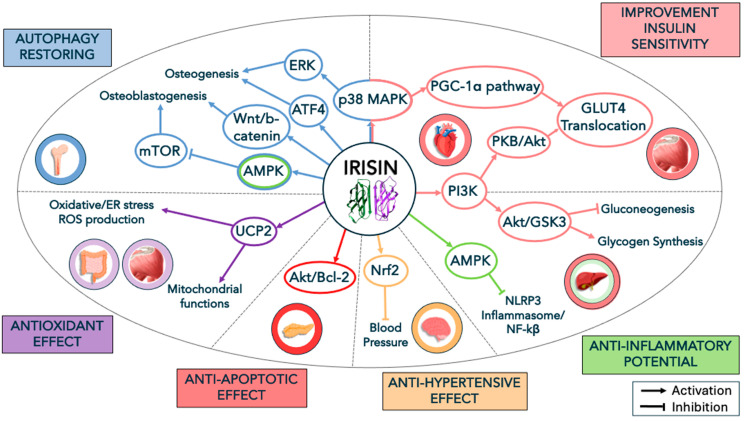
Irisin has pleiotropic effects, as it exerts multiple, diverse biological actions in different tissues and organ systems, meaning its effects are not limited to one specific function but span several physiological processes that impact metabolism, muscle function, fat storage, bone health, and brain activity. In particular, in skeletal muscle, irisin triggers the (MAPK)-PGC-1α pathway, thus improving oxidative phosphorylation and consequently mitochondrial respiration, together with an increase in GLUT4 translocation. At the same time, in cardiac tissue, GLUT4 translocation appears downstream of the activation of the PI3K/PKB/Akt pathway, with the final result of improving glucose uptake. In relationship to the liver, downstream, the PI3K trigger Akt/GSK3 is then activated, leading to gluconeogenesis reduction and the concomitant induction of glycogen synthesis. Altogether, with the stimulation of all the aforementioned pathways, irisin can promote sensitization to glucose in insulin-dependent tissue, like the heart, skeletal muscle, and the liver (**pink lines**). This adipo-myokine plays a crucial role in the hepatic metabolism, as AMPK phosphorylation exerts anti-inflammatory potential by decreasing NLRP3 inflammasome and NFKβ (**green lines**). Through AMPK phosphorylation, irisin also participates in autophagic-related mechanisms via mTOR inhibition promoting osteoblastogenesis by the parallel activation of Wtn/β catenin signaling. Furthermore, irisin is involved in osteogenesis involving ERK signaling via p38 MAPK, the same pathway mediating GLUT4 translocation in muscle tissue (**blue lines**). Irisin plays a crucial role not only in muscle–bone crosstalk but also acts on many other peripheral organs. Moreover, irisin shows antioxidant effects in the intestine and pancreas, as it can upregulate UCP2, reducing both oxidative stress and ER stress in intestinal cells, additionally increasing mitochondrial function and biogenesis (**violet lines**). Interestingly, irisin exerts a protective action on pancreatic β-cells to stimulate insulin synthesis and glucose-induced insulin secretion. Specifically, irisin can block the apoptotic effects induced by selective saturated fatty acids in β-cells via the Akt and Bcl-2 signaling pathway (**red lines**). Lastly, irisin acts as antihypertensive hormone, since it reduces blood pressure by activating the hypothalamic factor Nrf2, having a more selective effect on PVN nuclei for its antioxidant and anti-inflammatory action. Thus, this hormone cannot be simply identified for its single effects on specific organs, since its beneficial activity is provided by the synergistic combination of its action on multiple organs and systems.

**Figure 2 ijms-25-13480-f002:**
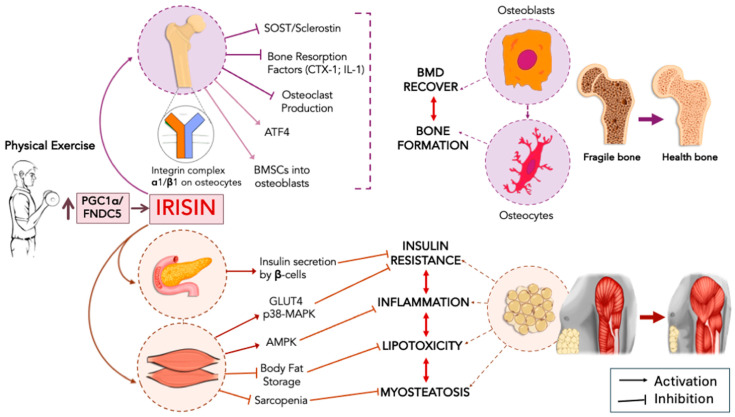
Skeletal muscle and bone are dynamic, adaptable tissues that function as endocrine and paracrine organs, maintaining tight reciprocal control and communication. Osteoporosis and sarcopenia often co-exist in obese patients with visceral adiposity, a condition known as osteosarcopenic obesity. This figure illustrates how irisin helps counteract and rebalance these pathological conditions by promoting muscle growth and bone formation. Physical exercise increases circulating irisin levels, as well as PGC-1α and FNDC5, leading to beneficial effects on bone health, particularly by stimulating pro-osteoblastic mechanisms and enhancing bone formation. Adiposity, characterized as a state of chronic low-grade inflammation, involves excessive lipid infiltration in skeletal muscle, which induces lipotoxicity and inflammation associated with myosteatosis. This fat deposition triggers the release of pro-inflammatory cytokines, affecting the osteoprotegerin pathway and promoting osteoclastogenesis and bone resorption. Irisin, by reducing inflammation and lipotoxicity (indicated by green arrows), can decrease bone resorptive factors, activate the Wnt/β-catenin signaling pathway, and stimulate osteoblast differentiation. Lipid accumulation also contributes to insulin resistance. Irisin reduces adipose tissue inflammation, improving the inflammatory impact of lipid deposits and lowering the risk of insulin resistance. Since insulin signaling deficiency is linked to alterations in bone microarchitecture, β-cell dysfunction is associated with osteoporosis, highlighting the direct relationship between insulin resistance and bone metabolism. Irisin acts as a hormonal messenger capable of counteracting insulin resistance and myosteatosis, thereby connecting insulin metabolism with muscle and bone homeostasis. Specifically, irisin can decrease NLRP3 inflammasome activity, promote GLUT4 translocation in muscle, and upregulate GLUT4 via the AMPK signaling pathway.

**Figure 3 ijms-25-13480-f003:**
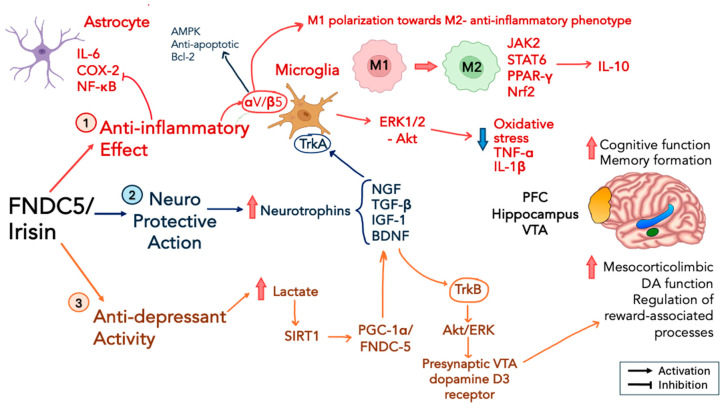
The figure depicts the capacity of irisin to affect multiple pathways involved in age-associated neurodegeneration and mechanisms underlying AD pathogenesis. Irisin serves as a crucial link between muscle contraction, myokine secretion, and brain function, embodying the concept of the muscle–brain axis. It has been shown to have potent (1) anti-inflammatory and (2) neuroprotective effects, such as protecting against Aβ neurotoxicity, reducing the release of pro-inflammatory cytokines, like IL-1β and IL-6, and inhibiting the expression of COX-2 and NF-κB in astrocytes. Irisin’s neuroprotective actions extend to preventing neuronal damage in conditions like middle cerebral artery occlusion and brain infarction, as well as mitigating microglial activation, neutrophil infiltration, and the expression of TNF-α and IL-6 via the ERK1/2 and Akt signaling pathways. Additionally, it counters the TLR4/MyD88-mediated neuroinflammatory response. The integrin αV/β5 receptor, which is highly expressed in microglia, plays a key role in irisin’s effects. Irisin has been shown to promote the shift from an M1 pro-inflammatory phenotype to an M2-like anti-inflammatory phenotype in microglia, leading to changes in their morphology. This shift is associated with an increase in AMPK phosphorylation and the expression of the anti-apoptotic protein Bcl-2. The interaction between irisin and BDNF signaling is also essential in neurodegenerative diseases, particularly Alzheimer’s disease (AD). For example, FNDC5 overexpression can mitigate the inhibitory effects of Aβ_1-42_ oligomers on BDNF expression. Exercise has been linked to increased (3) lactate levels and elevated hippocampal BDNF expression, with SIRT1 histone deacetylase activating the PGC-1α/FNDC5 signaling pathway to induce BDNF expression. Irisin’s (3) antidepressant potential is particularly relevant for managing neuropsychiatric symptoms in AD. Its ability to enhance dopaminergic activity (i.e., right side of the figure) may help address the pathophysiology of depression. Specifically, irisin-induced BDNF expression can activate dopamine D3 receptor-mediated signaling through the Akt and ERK pathways, fine-tuning dopamine release and contributing to a reduction in Aβ deposits and the overall Aβ burden.

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
