# Peer review of "Irisin: A Multifaceted Hormone Bridging Exercise and Disease Pathophysiology"

_ijms, 2024, doi:10.3390/ijms252413480_

Round 1
Reviewer 1 Report
Comments and Suggestions for Authors
Strengths of the Manuscript:
Comprehensive Scope: The manuscript extensively covers the physiological and pathological roles of irisin, including its effects on metabolic diseases, neurodegeneration, and musculoskeletal health. The breadth of topics ensures a holistic understanding.
Engagement with Current Literature: The authors cite numerous recent studies, demonstrating a strong grounding in the field.
Logical Flow: The paper transitions well between sections, moving from metabolic impacts to neurodegenerative diseases and finally summarizing its findings effectively.
Visual Aids: Figures and diagrams enhance the understanding of complex signaling pathways involving irisin.
Key Areas for Improvement:
1. Introduction and Contextualization
While the introduction provides an overview of irisin's discovery and its roles, it could better define gaps in the literature the review intends to address.
The hypothesis or objective of the narrative review should be explicitly stated.
2. Methodological Transparency
Although this is a narrative review, providing a structured methodology (e.g., databases searched, inclusion criteria) would improve reproducibility and robustness.
3. Critical Analysis
The review often summarizes findings without critiquing the methodologies or limitations of the cited studies. Including a critical evaluation would enhance its scholarly impact.
Contradictions in the literature, such as discrepancies in findings regarding irisin levels in health and disease, could be discussed more thoroughly.
4. Clarity and Conciseness
Certain sections, particularly those discussing molecular mechanisms, are overly detailed and dense, potentially overwhelming readers. Summarizing key findings while directing interested readers to specific studies would improve readability.
5. Visual Elements
While figures are included, some lack sufficient explanation or references in the main text. Each figure should be clearly integrated into the discussion.
6. Language and Grammar
The manuscript contains minor grammatical errors and awkward phrasing that detracts from its professional tone. A thorough proofreading is recommended.
7. Conclusions and Future Directions
The conclusions highlight irisin's therapeutic potential but lack specificity in proposing future research directions. For example, identifying gaps such as the lack of long-term human studies on recombinant irisin therapy would be valuable.
Recommendations for Revision:
Title: The title could be streamlined for clarity and impact. For example, "Irisin: A Multifaceted Hormone Bridging Exercise and Disease Pathophysiology."
Abstract: Make the abstract more concise by focusing on key findings and implications.
Figures: Enhance the figures with detailed captions explaining their relevance to the manuscript's discussion.
Critical Discussion: Add sections critiquing controversial findings or gaps in knowledge.
Technical Language: Simplify jargon-heavy sections for accessibility while retaining technical rigor.
Author Response
Strengths of the Manuscript:
Comprehensive Scope: The manuscript extensively covers the physiological and pathological roles of irisin, including its effects on metabolic diseases, neurodegeneration, and musculoskeletal health. The breadth of topics ensures a holistic understanding.
Engagement with Current Literature: The authors cite numerous recent studies, demonstrating a strong grounding in the field.
Logical Flow: The paper transitions well between sections, moving from metabolic impacts to neurodegenerative diseases and finally summarizing its findings effectively.
Visual Aids: Figures and diagrams enhance the understanding of complex signaling pathways involving irisin.
We would like to express our gratitude to the reviewer for his/her thorough and thoughtful evaluation of the manuscript, as well as for the positive feedback provided.
All modifications have been highlighted in red.
Key Areas for Improvement:
- Introduction and Contextualization
While the introduction provides an overview of irisin's discovery and its roles, it could better define gaps in the literature the review intends to address. The hypothesis or objective of the narrative review should be explicitly stated.
As suggested, we have added several lines (from lines 48 to 58) in the introduction to outline the aims and hypotheses of the review.
- Methodological Transparency
Although this is a narrative review, providing a structured methodology (e.g., databases searched, inclusion criteria) would improve reproducibility and robustness.
As suggested, we have added several lines (from lines 59 to 66) in the introduction to outline both methodology and search strategy of the review.
- Critical Analysis
The review often summarizes findings without critiquing the methodologies or limitations of the cited studies. Including a critical evaluation would enhance its scholarly impact. Contradictions in the literature, such as discrepancies in findings regarding irisin levels in health and disease, could be discussed more thoroughly.
To provide a comprehensive discussion of the main limitations in the findings, including methodological constraints, we have added a new section (Section 5, lines 780 to 843) where we present several examples of potential contradictions and discrepancies.
- Clarity and Conciseness
Certain sections, particularly those discussing molecular mechanisms, are overly detailed and dense, potentially overwhelming readers. Summarizing key findings while directing interested readers to specific studies would improve readability.
As suggested, to enhance clarity, fluency, and cohesion without compromising necessary complexity, a large number of sections or paragraphs have been completely rephrased.
- Visual Elements
While figures are included, some lack sufficient explanation or references in the main text. Each figure should be clearly integrated into the discussion.
As suggested, each figure is now recapped throughout the text (e.g., for figure 1, lines 142-143; line 163; line 193); (e.g., for figure 2, line 351; line 385; line 431; line 473); (e.g., for figure 3, line 620; line 682; line 741).
- Language and Grammar
The manuscript contains minor grammatical errors and awkward phrasing that detracts from its professional tone. A thorough proofreading is recommended.
The manuscript has undergone thorough and methodical proofreading. Many sentences have been rephrased where necessary, and all changes have been highlighted in red.
- Conclusions and Future Directions
The conclusions highlight irisin's therapeutic potential but lack specificity in proposing future research directions. For example, identifying gaps such as the lack of long-term human studies on recombinant irisin therapy would be valuable.
The last paragraph “conclusions” has been renamed as “conclusions and future directions”. Many lines (922-942) have been added to discuss future research directions in several fields, including gaps of knowledge linked to the lack of long-term human studies.
Recommendations for Revision:
- Title: The title could be streamlined for clarity and impact. For example, "Irisin: A Multifaceted Hormone Bridging Exercise and Disease Pathophysiology."
Following the proposal, the original title has been replaced with the one suggested.
- Abstract: Make the abstract more concise by focusing on key findings and implications.
The abstract has been considerably condensed focusing on the aspects suggested.
- Figures: Enhance the figures with detailed captions explaining their relevance to the manuscript's discussion.
Figure 1 has been edited de novo by inserting a new graphic for signals and clarifying the meaning of the different signals in terms of activation versus inhibition. Figure 2 has been entirely redesigned, with a new graphic for signals inserted to clarify the meaning of the different signals in terms of activation versus inhibition. The legend to figure 3 has been entirely rewritten.
- Critical Discussion: Add sections critiquing controversial findings or gaps in knowledge.
To provide a comprehensive discussion of the main limitations and controversial findings, including methodological constraints, we have added a new section (Section 5, lines 780 to 843) where we present several examples of potential contradictions and discrepancies. Moreover, several lines have been inserted to discuss controversial findings and possible explanations (e.g, lines 118-120 or 281-290).
- Technical Language: Simplify jargon-heavy sections for accessibility while retaining technical rigor.
As suggested, to enhance clarity, fluency, and cohesion without compromising necessary complexity, a large number of sections or paragraphs have been completely rephrased. The manuscript has been entirely and methodically proofread. When necessary many sentences have been reshaped. All changes have been highlighted in red.
Reviewer 2 Report
Comments and Suggestions for Authors
Current review article described the role of irisin in physiology and pathology in a good way. Please conduct the concerns below.
1. Receptor of irisin seems ignored. Why?
2. Figure 1 showed the effects of irisin in detail. However, the priority of each signal was not introduced.
3. Figure 2 mentioned the effects of irisin on bone formation and muscle growth. Signals will be helpful in each pathway.
4. It is hard to follow Figure 3, probably the legends were not enough.
5. Legends for Figure 3 must distinguish with discussion.
6. Limitation(s) of the article may strengthen it.
Author Response
Current review article described the role of irisin in physiology and pathology in a good way. Please conduct the concerns below.
- Receptor of irisin seems ignored. Why?
The irisin’s receptor, integrin αV/β5, has been described and cited (e.g., line 264). However, we implemented the information about this receptor by adding new lines (and discussion) between lines 266-272, as well as in line 598 and between 603 and 608. Also, in legend to figure 3 (line 762).
- Figure 1 showed the effects of irisin in detail. However, the priority of each signal was not introduced.
Figure 1 has been edited de novo by inserting a new graphic for signals and clarifying the meaning of the different signals in terms of activation versus inhibition.
- Figure 2 mentioned the effects of irisin on bone formation and muscle growth. Signals will be helpful in each pathway.
Figure 2 has been entirely redesigned, with a new graphic for signals inserted to clarify the meaning of the different signals in terms of activation versus inhibition.
- It is hard to follow Figure 3, probably the legends were not enough.
Ilaria
- Legends for Figure 3 must distinguish with discussion.
The legend to figure 3 has been entirely rewritten, as suggested.
- Limitation(s) of the article may strengthen it.
To provide a comprehensive discussion of the main limitations in the findings, including methodological constraints, we have added a new section (Section 5, lines 780 to 843) where we present several examples of potential contradictions and discrepancies.
Moreover, the last paragraph “conclusions” has been renamed as “conclusions and future directions”. Many lines (922-942) have been added to discuss future research directions in several fields, including gaps of knowledge (and limitations) linked to the lack of long-term human studies.